# Assessing nanobody interaction with SARS-CoV-2 Nsp9

Gennaro Esposito [1,2]*, Yamanappa Hunashal[1], Mathias Percipalle[1¤],
Federico Fogolari[2,3], Tomas Venit[1], Ainars Leonchiks[4], Kristin C. Gunsalus[5,6],
Fabio Piano[5,6], Piergiorgio Percipalle[1,6]*

**1** Division of Science, New York University Abu Dhabi, Abu Dhabi, UAE, **2** Istituto Nazionale Biostrutture e Biosistemi, Roma, Italy, **3** Dipartimento di Scienze Matematiche, Informatiche e Fisiche, Università di Udine, Udine, Italy, **4** ASLA Biotech AB, Riga, Latvia, **5** Department of Biology and Center Genomics System Biology, NYU, New York, New York, United States of America, **6** Center Genomics System Biology, New York University Abu Dhabi, Abu Dhabi, UAE

¤ Current address: Institute of Science and Technology Austria, Klosterneuburg, Austria
* rino.esposito@nyu.edu (GE); pp69@nyu.edu (PP)

**Data Availability Statement:** The NMR spectra files are available from the Dryad repository and are accessible with DOI: 10.5061/dryad.905qfttsj.

**Funding:** The authors received no specific funding for this work. YH and MP worked on the NYUAD

## Abstract

The interaction between SARS-CoV-2 non-structural protein Nsp9 and the nanobody 2NSP90 was investigated by NMR spectroscopy using the paramagnetic perturbation methodology PENELOP (Paramagnetic Equilibrium vs Nonequilibrium magnetization Enhancement or LOss Perturbation). The Nsp9 monomer is an essential component of the replication and transcription complex (RTC) that reproduces the viral gRNA for subsequent propagation. Therefore preventing Nsp9 recruitment in RTC would represent an efficient antiviral strategy that could be applied to different coronaviruses, given the Nsp9 relative invariance. The NMR results were consistent with a previous characterization suggesting a 4:4 Nsp9-to-nanobody stoichiometry with the occurrence of two epitope pairs on each of the Nsp9 units that establish the inter-dimer contacts of Nsp9 tetramer. The oligomerization state of Nsp9 was also analyzed by molecular dynamics simulations and both dimers and tetramers resulted plausible. A different distribution of the mapped epitopes on the tetramer surface with respect to the former 4:4 complex could also be possible, as well as different stoichiometries of the Nsp9-nanobody assemblies such as the 2:2 stoichiometry suggested by the recent crystal structure of the Nsp9 complex with 2NSP23 (PDB ID: 8dqu), a nanobody exhibiting essentially the same affinity as 2NSP90. The experimental NMR evidence, however, ruled out the occurrence in liquid state of the relevant Nsp9 conformational change observed in the same crystal structure.

## Introduction

Most available SARS-CoV-2 vaccines are targeted against the spike (S) protein [1,2]. However, its high rate of mutation presents challenges to the continued development of effective vaccines: in fact, S and the nucleocapsid (N) protein have shown the highest mutation frequency among the structural proteins of the virus, whereas the transmembrane (M) and the envelope

institutional grant of GE. TV worked on the NYUAD institutional of PP.

**Competing interests:** GE and PP are coauthors of the worldwide patent WO2023041985A2 filed by New York University in Abu Dhabi.

(E) proteins exhibit a relatively lower number of mutations [3]. Although vaccines directed against the whole SARS-CoV-2 viral particle might mitigate the effects of S protein mutation by eliciting a wider repertoire of immune responses, alternative routes can also be explored to enrich the therapy arsenal [4]. These include repurposed and newly designed small-molecule antivirals that mostly target SARS-CoV-2 non-structural proteins (Nsps) [4,5], as well as several types of antiviral antibodies—such as monoclonal, nanoparticle-conjugated and cocktail antibodies—that block the virus binding site to the host or modulate the inflammatory response [5]. Among these, the most advantageous protein-based antivirals are nanobodies, which correspond to the antigen binding domains of heavy-chain-only antibodies that naturally occur in camelids and sharks [6]. Due to their small size, high solubility and folding stability, nanobodies are at the forefront of the evolving field of antibody therapeutics [7]. Consequently, nanobodies have the potential to circumvent challenges of production, conservation, handling and administration commonly associated with antibody therapeutics.

The vast majority of reported anti-SARS-CoV-2 nanobodies were raised against the receptor binding domain of S protein, but different strategies such as humanization of frame regions, fusion with the IgG1-Fc domain, and combinations of different nanobody clones were necessary to achieve satisfactory neutralizing effects *in vitro* [reviewed in 8–10]. Some nanobodies were also raised to target the N protein, though only for diagnostic purposes [8]. Among the SARS-CoV-2 non-structural proteins, only Nsp3 [11] and Nsp9 [12] were considered for nanobody challenge. Nsp3 includes the papain-like protease domain, one of the two self-activating proteases that are crucial for the virus mechanism of action to cleave the polyproteins translated after the host infection [13]. The Nsp9 monomer is an essential component of the replication and transcription complex (RTC) [14] that reproduces the viral gRNA for subsequent propagation [13,14].

Several factors make SARS-CoV-2 Nsps attractive targets for the development of new antiviral nanobodies that bind and interfere with the viral transcription, replication and propagation machinery. First, only 45% of all Nsp sequence sites display mutations, in contrast with 96.5% for the S protein [15,16]. Of the 16 Nsps, several exhibit quite low mutation propensity, with Nsp7, Nsp8, Nsp9 and Nsp10 accounting for 5.3% of the total protein sequences (~ 1% for Nsp9), but contributing less than 1% (0.1% for Nsp9) of total mutations occurring in the entire viral genome [15]. Moreover, humoral immunity mapping has shown that most of the binding paratopes of IgM and IgG recognize SARS-CoV-2 Nsps, indicating that these are effective triggers of endogenous immune responses [17].

Therefore, to explore the possibility of a novel antiviral strategy, we set out to raise nanobodies against SARS-CoV-2 Nsp9. Out of 136 different nanobodies identified by enzyme-linked immunosorbent assay (ELISA) tests, the best performing ones (2NSP23 and 2NSP90) were shown to be equivalent and to recognize both recombinant and endogenous Nsp9 in the saliva of SARS-CoV-2 infected patients [12]. Characterization of Nsp9 by NMR upon titration with both nanobodies demonstrated that the signal evolution was consistent with a tetrameric assembly of Nsp9, which could impair its recruitment as a monomer in the RTC [13,14]. The analysis also enabled us to locate Nsp9 epitopes on the surface of the tetramer [12]. A subsequent NMR study on isolated Nsp9 further supported the occurrence of a composite oligomerization pattern, with Nsp9 monomers, dimers and tetramers exchanging in the absence of nanobodies [18].

We report here further characterization of the interaction between Nsp9 and the 2NSP90 nanobody using an NMR paramagnetic perturbation methodology set up in our laboratory [18–20]. The oligomerization state of Nsp9 is also evaluated by molecular dynamics simulations. While confirming the identification of the previous epitopes [12], their different distribution on the tetramer surface with respect to the former proposal could also be considered, as

well as different stoichiometries of the Nsp9-nanobody assemblies as suggested by a recently published crystal structure of the Nsp9-2NSP23 complex [21].

## Materials and methods

### Protein expression and purification

The uniformly $^{15}N,^{13}C$-labeled SARS-CoV-2 Nsp9 was expressed and purified by ASLA Ltd. (Riga, Latvia) according to the protocol previously described [12] leading to a final sequence with an additional GlyAlaMetGly tetrapeptide at the N-terminus. The same recipe was employed to express the unlabeled SARS-CoV-2 Nsp9 employed for mass photometry. For brevity, the double labeling notation of the Nsp9 samples used for electron spin resonance (ESR) and NMR spectra will be omitted in the following. The unlabeled 2NSP90 was expressed according to the standard procedure reported for all nanobodies raised against Nsp9 [12] and purified by immobilized metal affinity chromatography exploiting the C-terminal $(His)_6$ tag.

### NMR and ESR spectroscopy

The samples for paramagnetic perturbation experiments were prepared in $H_2O/D_2O$ 95/5 (vol/vol), 10 mM phosphate buffer, 150 mM NaCl, 0.004% $NaN_3$ (Sigma), 0.4 mM TCEP (tris (2-carboxyethyl)phosphine) (Sigma), pH 7.1. The Nsp9 concentration was always 18 μM, both in the absence and presence of 5.6 mM 2NSP90. The protein concentrations were determined by UV absorption at 280 nm. A mother solution of 1.0 M TEMPOL (4-hydroxy-2,2,6,6-tetra-methyl-piperidine-l-oxyl) (Sigma) was prepared in $D_2O$ and stored in fridge with light-protection wrapping. Microliter aliquots of that preparation were added to the protein solutions to reach 1mM nitroxide.

The NMR experiments were carried out at 14.0 T (600.19 MHz $^1H$ frequency, 60.82 MHz $^{15}N$ frequency) and 298 K on a Bruker Avance III spectrometer equipped with a triple reso-nance, z-axis gradient cryoprobe. Two-dimensional $^{15}N$-$^1H$ HSQC experiments were acquired via double INEPT transfer [22], sensitivity-improved Echo/Antiecho-TPPI pure phase detec-tion in $F_1$ [23], gradient coherence selection [24], and flip-back pulse for solvent suppression [25]. The spectra were recorded over spectral widths of 40 ppm ($^{15}N$, $t_1$) and 14 ppm ($^1H$, $t_2$) and digitized over 80–128 and 2048 points, respectively. Different relaxation delays, namely 3 and 0.3 s, were set to collect the equilibrium and off-equilibrium data, respectively, according to the PENELOP (Paramagnetic Equilibrium vs Nonequilibrium magnetization Enhancement or LOss Perturbation) protocol [18]. Data were processed with TOPSPIN version 4.0.6. Prior to Fourier transformation, linear prediction in $t_1$ (up to 192 points) and zero filling were applied to yield the final data set of 2 K x 1 K points. All resonance assignments were from Bio-logical Magnetic Resonance Bank (BMRB 50513) and Buchko et al. [26]. The NMR signal attenuations and the relative errors were calculated according to the protocol previously reported [18] (data in S1 File).

To safely apply the PENELOP protocol, control ESR spectra were acquired to check the invariance of TEMPOL rotational correlation time ($\tau_c$), in the presence of either Nsp9 alone or Nsp9 and nanobody, in order to rule out specific interactions of the nitroxide with the proteins [18,20].

The ESR experiments were performed with a Bruker EMXnano spectrometer operating in the X band, at ambient temperature and different TEMPOL/protein(s) molar ratios, namely, 1:0(:0) and 1:1(:1). TEMPOL solutions (10 μM) were prepared in 10 mM phosphate, 150 mM NaCl, 0.8 mM TCEP, pH 7.1, without and with proteins at the above mentioned ratios. Capillaries filled with 50 μL of each solution were placed in standard 4 mm tubes and submit-ted to acquisition. The ESR operating parameters were: frequency = 9.6 GHz; microwave

power = 0.316 mW; modulation amplitude = 1 Gauss; modulation frequency = 100 kHz; center field = 3429.8 Gauss; sweep width = 200 Gauss; time constant = 1.28 ms. The data were processed using the Bruker software package Xenon and the $\tau_c$ values determined as previously described [18–20].

## Molecular simulations

**Molecular dynamics.** The tetramer and dimer of SARS-CoV-2 Nsp9 were taken from the structures deposited in the Protein Data Bank (PDB ID 7bwq and 6w4b, respectively). The tetramer was excised from the asymmetric unit entailing six subunits. The monomer was excised from the dimer. The structures were soaked in a box of TIP3P water [27] and 0.150 M NaCl up to at least 14 Å from any solute atom using the program VMD [28]. All molecular simulations were performed using the program NAMD2 [29]. First each system was energy minimized by 5,000 steepest descent steps. The dynamics was started at 0 K and the temperature was increased to the target temperature in 10 ps and equilibrated for 1 ns rescaling the temperature every 0.1 ps. During this phase and in all simulations, pressure was kept constant at 1.0 atmosphere by the Langevin piston method with period 200 ps and decay time 100 ps, at the target temperature [30,31]. In all simulations, except for the heating phase described above, the temperature was controlled through Langevin dynamics with damping constant 2.5 ps$^{-1}$. Interactions were gradually switched off at 12 Å starting at 10 Å. The time step was 1 fs for bonded interactions and 2 fs for nonbonded interactions. Hydrogen bond lengths were restrained by the algorithm Settle [32]. Simulations were performed for 100 ns.

**Enthalpy calculations.** The forcefield energy recorded during the simulations provides a good approximation to the enthalpy for condensed states undergoing minimal volume variations upon dimerization or tetramerization. In order to compare simulations with different numbers of $H_2O$, $Na^+$ and $Cl^-$, simulations were performed employing exactly the same forcefield and simulation protocol adjustable parameters for pure water, water with $Na^+$ and $Cl^-$ ions in equal number and slightly different number ($< 10\%$). The average molar energies for $H_2O$, $Na^+$ and $Cl^-$, computed with the same parameters and ionic strength used for all simulations, were estimated as −9.98, −82.06 and −89.53 kcal/mol, respectively. Expressing the average energy of the system as the sum of terms due to each water and each ion (in the condition of ~0.15 M salt concentration) it was possible to correct for the different numbers of waters and ions in the different simulations.

**Entropy calculations.** The conformational freedom reduction upon monomer-monomer and dimer-dimer association was estimated using the program PDB2ENTROPY [33], which computes the entropy based on the k-th nearest neighbour method in the space of torsional angles. Correlation among different torsions was taken into account using the Maximum Information Spanning Tree method as implemented in the same program [33]. The association entropy was computed using the program PDB2TRENT [33], which computes the entropy based on the k-th nearest neighbour method in the six-dimensional space of rotation/translations. First one of the two groups of associating atoms (e.g. one of the dimers) is used for superposition and then the positional/orientational entropy of the other is computed.

**Solvation energy calculations.** The solvation energy was computed using the Generalized Born / Surface Area model as implemented in the program Bluues2 [34]. The internal dielectric constant was set to 4 and the ionic strength was set to 0.15 M. The surface tension coefficient was set to 5 cal/mol A$^2$. The computed free energies strongly depend on the chosen parameters, but provide overall trends upon comparison.

**Umbrella sampling simulations.** A collective variable entailing the α-carbons of the subunits to be dissociated was created and progressively forced to remove the interaction between

the dissociating subunits. NAMD2 [29] implementation of Umbrella Sampling [35] was used to report the accumulated work as the corresponding free energy. The simulation lasted 50 ns. Due to the short time, the estimated work is expected to be larger than the actual free energy of the process. The difference between monomer-monomer and dimer-dimer association free energy, rather than the absolute values, is considered for qualitative comparison of the two processes.

## Results

### PENELOP-based epitope mapping

To apply the PENELOP method [18], protein NMR experiments are acquired under equilibrium (long recovery delay) and off-equilibrium (short recovery delay) conditions. For each condition, two spectra are recorded, in the absence and presence of a paramagnetic species, respectively. The paramagnetic species can be considered to randomly sample the accessible surface of the probed molecule when strong specific interactions can be excluded [36–38]. The long-recovery-delay spectra report the equilibrium paramagnetic perturbation and map the surface accessibility of the protein in terms of signal intensity reduction expressed as normalized attenuations, $A_N[eq]$ [18–20,36] (data in S1 File). Values of $A_N[eq] > 1$, i.e. above the average perturbation, characterize surface exposure [18–20,36]. The short-recovery-delay spectra report the off-equilibrium perturbation expressed by $A_N[off]$, corresponding to the normalized attenuations measured for a particular nonequilibrium steady state of the signal magnetizations that depends on the chosen recovery delay. Compared to the corresponding equilibrium values, the normalized off-equilibrium attenuations can be either larger, i.e. $A_N[off] > A_N[eq]$ defined Type I pattern, or smaller, i.e. $A_N[off] < A_N[eq]$ defined Type II pattern [18–20], as summarized in the scheme of Fig 1.

The Type I pattern represents signals with slower recovery under fast recycling conditions, with respect to the equilibrium acquisitions, and reveals limited accessibility of the paramagnetic probe to the corresponding molecular locations due to hindered surface or structural burial. On the other hand, the Type II pattern identifies signals with faster recovery in fast recycling experiments compared to equilibrium acquisitions. This can occur because of

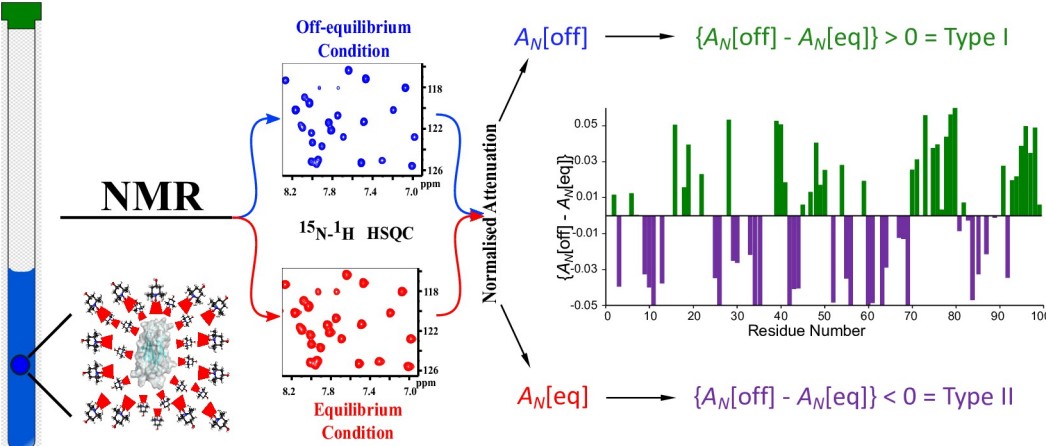

**Fig 1. PENELOP protocol.** The scheme summarizes the workflow of the protocol for the acquisition of the experimental data under conditions of equilibrium and non-equilibrium, and the subsequent elaboration leading to the definition of Type I and Type II pattern. Type I pattern identifies sites with hampered accessibility. Type II pattern identifies sites of conformational or chemical exchange [18–20].

efficient relaxation at the specific molecular location originating from an exchange process occurring over μs−ms time scale due to conformational or interaction/association dynamics.

A preliminary ESR control to measure the TEMPOL $\tau_c$ in solution, when alone, with Nsp9, and with Nsp9 and 2NSP90 or 2NSP23 nanobodies gave 30, 28, 30 and 26 ps, respectively (S1 Fig). These values, which are equal within the experimental uncertainty (10%), confirm the previously observed ones [18,20] and the absence of relevant specific interactions between the nitroxide and the proteins, thereby ensuring validity for a paramagnetic perturbation analysis by PENELOP approach [18].

Fig 2 portrays the overlay of the Nsp9 $^{15}$N−$^{1}$H HSCQ spectra collected under equilibrium conditions without and with TEMPOL, in the absence (Fig 2A) and presence (Fig 2B) of 2NSP90. The corresponding off-equilibrium spectra are reported in S2 Fig. A clear reduction of red overlay can be readily appreciated in panel 2B, suggesting that the relative uniformity of attenuation induced by the nitroxide is reduced in the presence of the nanobody, consistently with the increase of Nsp9 surface burial from the expected interaction.

This intuitively reasonable conclusion can be refined by applying the PENELOP method [18]. Fig 3 displays the result of the analysis. The histograms depict the differences between off-equilibrium and equilibrium $A_N$ values of the individual amide signals of the protein, obtained from $^{15}$N−$^{1}$H HSCQ spectra with and without TEMPOL. The original $A_N$ data are reported in S3 Fig.

The bar plots in Fig 3 highlight the locations of the Type I pattern (green bars) and Type II pattern (purple bars) along Nsp9 sequence, in the absence (Fig 3A) and presence (Fig 3B) of 2NSP90 nanobody. The chromatic code of these patterns is changed in correspondence of the Nsp9 epitopes targeted by 2NSP90 (and 2NSP23), and previously identified from Nsp9 signal attenuations and losses [12]. In particular, in Fig 3, orange, red, yellow and sky-blue highlight epitopes e1 (Q11, M12, S13, C14, L29), e2 (K86, L45, S46, N27), e3 (D50, L51, K52, W53) and e4 (C73, R74, F75, V76, Y87, L88, Y89), respectively, all pictured in Fig 4.

That epitope identification was achieved after excluding the dimer and tetramer interfaces, along with adjacent segments or single residues (data in S1 Table), thereby implying a tetrameric Nsp9 assembly [39]. It is worth pointing out that the paramagnetic perturbation measurements here described for applying the PENELOP protocol to epitope mapping in the Nsp9-2NSP90 system were performed at protein concentration ratio below the occurrence of any NH cross-peak loss, based on the previous observations [12]. The comparison of the bar graphs in the two panels of Fig 3 shows that the presence of 2NSP90 tends to shift the Nsp9 $A_N$ pattern of the epitope regions from Type II towards Type I, i.e. from exchange dynamics to hampered accessibility. This is not the case for a large part of epitope e1, that instead exhibits an increased contribution from exchange dynamics (Type II pattern). The occurrence of Nsp9-2NSP90 binding interactions is expected to hinder TEMPOL accessibility at the involved surfaces of both proteins, which is consistent with the observed shift towards Type I pattern of Nsp9 $A_N$ values. The Type II pattern increasing that characterizes 2NSP90 challenge at Nsp9 epitope e1 may suggest a weaker binding at that site (stressed by the low nanobody/Nsp9 ratio), translating into an additional exchange which enhances the Type II pattern response with respect to the isolated Nsp9. This interpretation agrees with the effects that are seen at the edges of epitopes e3 and e4 where weak Type II patterns establish (Fig 3). Overall, the salient feature emerging from the PENELOP-based epitope mapping of Nsp9 is a compelling analogy with the identification obtained from the earlier titrations with 2NSP90 and 2NSP23 nanobodies that were interpreted based on progressive loss and attenuation speed of the backbone amide NMR signals and the chemical shift evolution at the N-terminal and C-terminal residues with increasing nanobody concentrations [12], coupled to the structural restraints of the different oligomeric stoichiometries available from the crystal structures of Nsp9 [39–45].

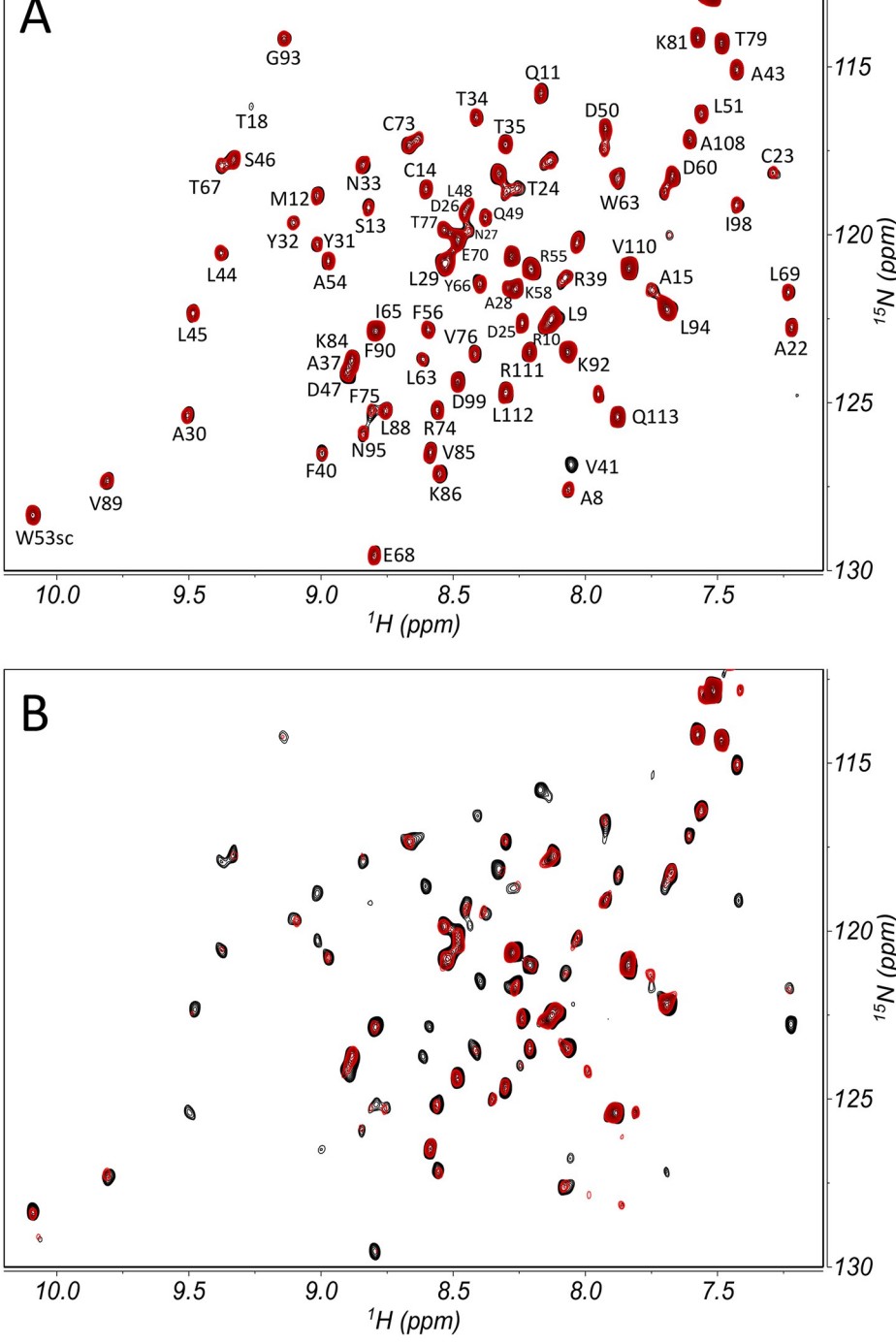

**Fig 2. Paramagnetic perturbation of Nsp9 spectra.** $^{15}$N–$^{1}$H HSCQ spectra overlay of labeled SARS-CoV-2 Nsp9 without (A) and with (B) unlabeled 2NSP90, in the absence (black contours) and presence (red contours) of TEMPOL. The assignments of the backbone NHs (and W53 side chain) [26] are also reported. Data collection was carried out at 298 K on 18 µM Nsp9 alone or with 5.6 µM 2NSP90, before and after addition of 1 mM TEMPOL, with a relaxation delay of 3 s, according to the PENELOP protocol for equilibrium acquisitions [18]. The contour plot pairs with and without TEMPOL are always drawn at the same vertical scale, whereas a two-fold scale increment is applied to panel B data (with 2NSP90) compared to panel A data (without 2NSP90).

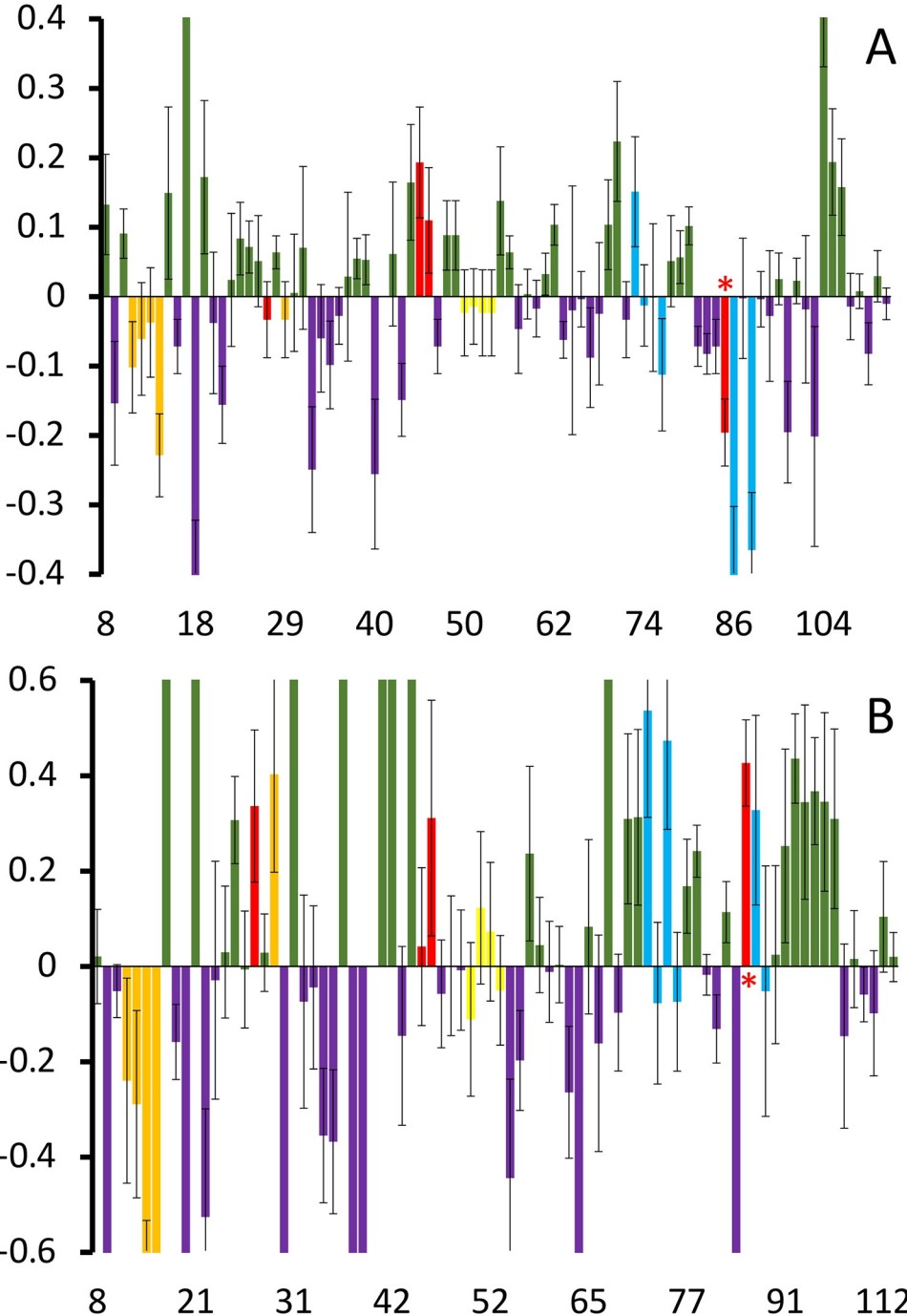

**Fig 3. Nsp9 normalized attenuation differences.** Bar plot of $\{A_N[\text{off}] - A_N[\text{eq}]\}$ differences highlighting the locations of the Type I pattern (green bars) and Type II pattern (purple bars) for 18 µM SARS-CoV-2 Nsp9 alone (A) and in the presence of 5.6 µM 2NSP90 (B). Both plots are truncated to highlight the variations at the epitope locations. Different colors indicate the positions of the Nsp9 epitopes inferred from nanobody titration experiments [12] (orange = e1; red = e2; yellow = e3; sky-blue = e4 –see text). The data for segments and residues 1–7, 20, 36, 59, 87, 96–106 and the relative abscissa points are not reported because of the absence of the corresponding signals in one or more spectra. Because of the lack of NH, also proline locations are skipped on the abscissa axis. The inherently weak signals of Y87 (e4) turned out to be unobservable when TEMPOL was added, irrespective of the nanobody presence. Therefore the signal of V85 (with red asterisk) is considered in the epitope highlight scheme to obtain some clue on the edge region between epitopes e2 and e4. The vertical scales have been expanded to highlight the details. The full plots without truncation are given in S4 Fig.

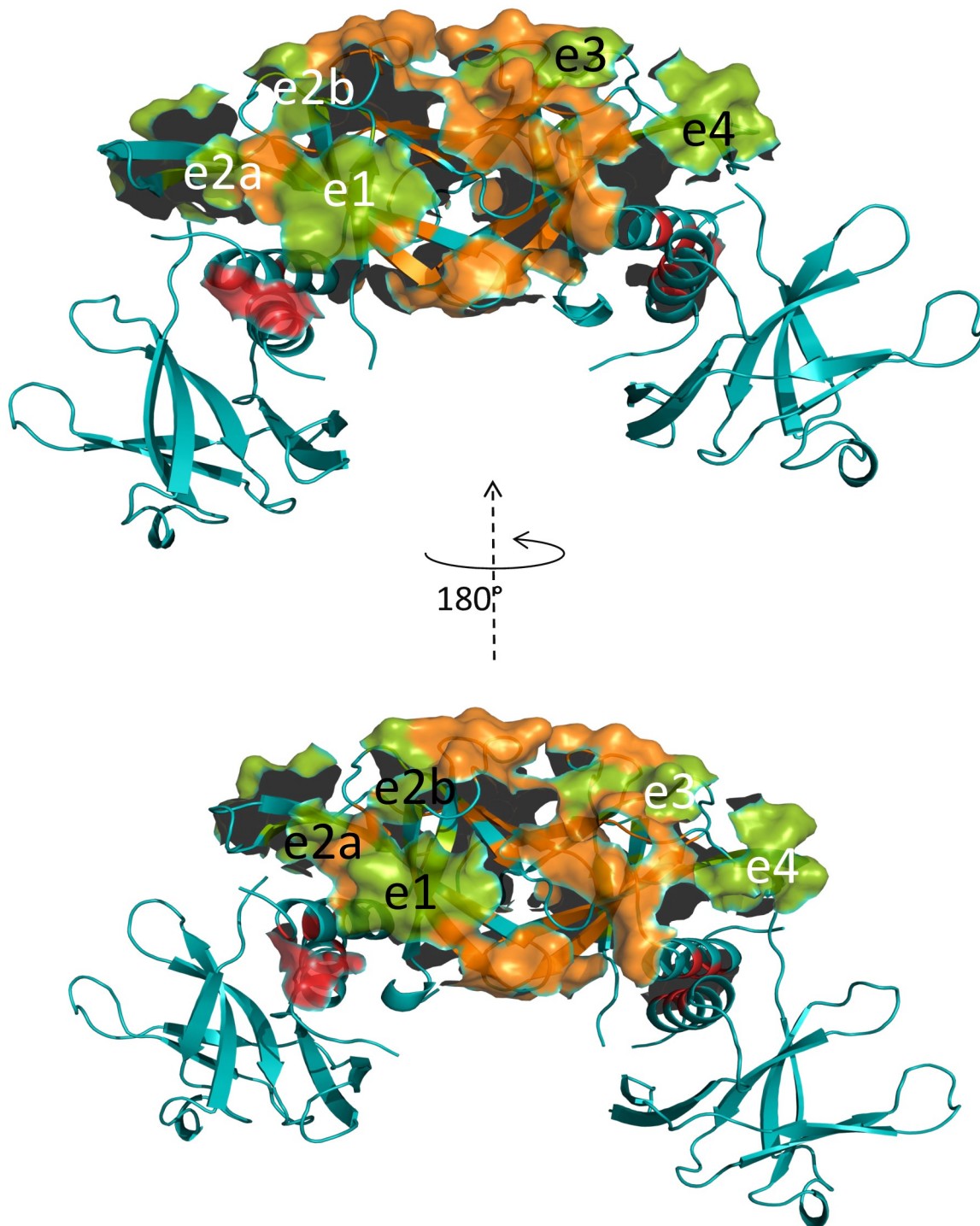

**Fig 4. Epitopes of Nsp9 interacting with nanobodies.** The SARS-CoV-2 Nsp9 tetramer [39] with the green surfaces indicating the proposed locations of the epitopes interacting with nanobodies 2NSP90 and 2NSP23, as inferred from titration NMR experiments [12]. The interdimer and intradimer contact surfaces are highlighted in orange and red, respectively. In the previous NMR-based model [12], a nanobody molecule should contact the surface including the epitopes e1 (Q11-M12-S13-C14-L29) and e2 (K86-L45-S46-N27). The latter epitope is split into e2a and e2b to stress the contribution of L45 and S46 also involved in the tetramerization interface. A second nanobody molecule should contact the surface formed by epitopes e3 (D50-L51-K52-W53) and e4 (C73-R74-F75-V76-Y87-L88-Y89). An analogous epitope ensemble is present on the opposite face of the tetramer for the binding of two additional nanobody units, as shown by the 180˚-rotated structure, leading to a 4:4 stoichiometry for the complex. The labels of the four epitopes that are located on each central subunit are colored accordingly. The structure (PDB ID: 7bwq) was drawn with PyMol (Schroedinger LLC).

## Thermodynamic inferences

The average force field energy for the simulations of the monomer, the dimer and the tetramer were estimated after correction for the different numbers of $H_2O$ molecules and $Na^+$ and $Cl^-$ ions.

After the corrections, the average energy for the monomer to dimer and dimer to tetramer transitions were found to be $-14.5 \pm 1.3$ kcal/mol and $-151.4 \pm 1.8$ kcal/mol, respectively. The large difference in enthalpy between dimerization and tetramerization matches a similar difference in the change of interaction with solvent, i.e. 502.3 kcal/mol for dimerization and 337 kcal/mol for dimer-dimer association, as calculated from the same simulations. Because of the dependence of the $\Delta\Delta H$ between the processes on the interaction with the solvent, the changes in solvation free energy, which also includes the entropy of the solvent, were examined. One hundred snapshots were considered from all simulations, and the changes in solvation free energy were computed by the difference of the averages, using the Generalized Born / Surface Area model as implemented in Bluues2 [34]. Notwithstanding all the limitations of the approach, the computed solvation free energy, which implicitly takes into account also the entropy of water and salts, is more unfavorable for dimer-dimer association with respect to monomer-monomer association by $-46.2$ kcal/mol ($-97.3$ vs. $-51.1$ kcal/mol). The conformational entropy reduction calculated over 1,000 snapshots is more unfavorable for the dimer-dimer association with respect to monomer-monomer association by $-5.4$ kcal/mol ($-26.1$ kcal/mol vs. $-20.7$ kcal/mol at 310 K). The positional/orientational entropy is more unfavorable for the monomer-monomer association with respect to dimer-dimer association by $-1.7$ kcal/mol ($-8.9$ kcal/mol vs. $-7.2$ kcal/mol at 310 K). All these calculations suggest that dimer-dimer association is less favorable than monomer-monomer association due to solvation. Finally a simulation using umbrella sampling to force tetramer dissociation into dimers and dimer dissociation into monomers taking the centers of the α-carbons as collective variable, resulted in a difference of 15.8 kcal/mol more favorable for the monomer-monomer association ($-84.8$ kcal/mol) versus the dimer-dimer association ($-69.0$ kcal/mol). These figures are overestimating the real binding free energies due to the limited simulation time (50 ns), but the kind of errors implied are similar in the two simulations and therefore the difference in free energy should be in the correct range.

## Discussion

### Nsp9-nanobody complex stoichiometry

According to our previous NMR result interpretation, the interaction between SARS-CoV-2 Nsp9 and nanobodies 2NSP90 and 2NSP23 leads to complexes with 4:4 stoichiometry, and an apparent $\Delta G$ between $-4.9$ and $-6.8$ kcal/mol for the binding of each single nanobody unit to Nsp9 tetramer [12]. The structure of Nsp9 from coronaviruses has been widely addressed [39–45] and though most of the crystallographic evidence support a dimeric assembly through contacts between the C-terminal helices of the monomers, some heterogeneity has emerged. In particular, alternative inter-monomer interfaces involve β-sheet–β-sheet [40,44] and loop–β-sheet [41] contacts, along with tetrameric or hexameric stoichiometries of the crystal unit cells [40,41]. The oligomerization heterogeneity was also observed in solution, where tetramers and higher oligomers, in addition to the predominant dimers, were detected by dynamic light scattering and glutaraldehyde cross-linking [40]. The dimerization through the C-terminal helices was however deemed essential for viral replication [41] and RNA binding [45], given the loss of those functions for all the mutations of the contacting helix fragment. The elucidation of the RTC structural organization [14], where a single Nsp9 molecule participates through the same

helical fragment involved in the dimer interface, corrects and explains the wrong conclusion on the necessity of dimerization which, however, may be necessary for other cellular functions. For SARS-CoV-2 Nsp9, a tetrameric crystal structure was reported [39] entailing both inter-helix and inter-β-sheet interfaces (Fig 4), in addition to the canonical dimeric form [42]. In solution, a unexpected pattern for a 113-residue long protein was observed by three different NMR studies [12,26,46]. The spectra were always devoid of the signals from segments 1–7 and 97–106 and presented severe attenuation of several other cross-peaks, along with very poor extent of coherence transfer in 3D experiments. Considering that the missing segments match the dimer interface, a monomer-dimer exchange, intermediate on the NMR chemical shift time scale, was proposed which could explain intensity attenuations and defective 3D data, as well as the measured diffusion coefficient [12]. An exchange in fact broadens the NMR signal linewidths of the exchanging species when its frequency matches the chemical shift differences of the involved species. The broadening leads to flattening of the signals that merge with the baseline and become largely undetectable. Successively, dynamic association of the protein into a tetramer was inferred in solution too from the consistency of the paramagnetic pertur-bation pattern with the features of the tetramer crystal structure [18]. The dimerization con-stant reported from analytical ultracentrifugation determinations on SARS-CoV Nsp9 (identical to SARS-CoV-2 protein for all but 2 residues) corresponds to $\Delta G = -5.2$ kcal/mol [44]. This is an underestimated absolute value (by some 3 kcal/mol, at least) compared to gel filtration evidence [41]. According to the computed MD trajectories, the corresponding $\Delta H$ for SARS-CoV-2 Nsp9 dimer is $-14.5$ kcal/mol, in close agreement with another recent calcu-lation [47], whereas the same simulation protocol gives $\Delta H = -151.4$ kcal/mol for the tetramer of Fig 4, which makes conceivable its occurrence in solution. The difficulty of evaluating all the entropy contributions prevents a safe computation of $\Delta G$ values, but the above illustrated esti-mates of solvation, conformational and positional entropy definitely disfavor less (by ~ 50 kcal/mol) the dimer formation, in line with an umbrella sampling evaluation of the association processes that provides a $\Delta\Delta G \approx -30$ kcal/mol in favor of the dimer over the tetramer. Accord-ingly, the low concentration of the higher oligomers hinders the experimental assessment of the corresponding $K_d$ values. Resorting to alternative, highly sensitive techniques such as mass photometry does not appear feasible for molecules as small as Nsp9 and derived complexes under exchange conditions. Mass photometry can be safely applied to systems between 40–50 kDa and 10 MDa [48–50]. Therefore we exploited the PENELOP approach that had proven sufficiently sensitive to detect the tracks of the tetramer with 18 μM Nsp9 [18]. As shown in Fig 3, three of the four epitopes, that were already mapped by progressive peak loss and attenu-ation rate, exhibit the expected pattern change towards hindered accessibility upon titration with 2NSP90. At variance, about all of the residues of the N-terminal epitope (e1, Fig 4) display a significant increase of the Type II pattern, thus featuring an additional exchange contribution due to the nanobody interaction. This interpretation implicitly assumes that the dissociation rate of the nanobody paratope from the involved Nsp9 region, i.e. Q11, M12, S13, C14, while matching closely the limiting chemical shift difference between the free and bound states at those epitope locations, is also loose enough to preserve the Nsp9 surface accessibility for the paramagnetic probe. In other words, 2NSP90 binding to e1 epitope is weaker than the interac-tion at the other three epitope sites. Fig 3 also shows a clear increment of Type I pattern at frag-ment 90–95 in the Nsp9-2NSP90 system with respect to isolated Nsp9. Fragment 90–95 maps to an interdimer interface region involved in the Nsp9 tetramer assembly. It is very likely that the pattern observed at segment 90–95 represents a further tightening of the Nsp9 tetramer rather than an extension of the adjacent epitope e4 (Fig 4). All the nanobodies recognizing Nsp9, in fact, derive from Llama antibodies raised against a mutant of the protein more prone to aggregation after the substitution of the three Cys residues of the natural sequence with

three Ser residues [12]. Therefore, it is not surprising that the selected nanobodies tend to stabilize the higher oligomeric forms of Nsp9. This feature is crucial for possible antiviral properties of the anti-Nsp9 nanobodies that could limit the available concentration of Nsp9 monomers and therefore impair the RTC formation.

## An alternative model

An intriguing alternative model of Nsp9-nanobody complex was recently published [21] involving nanobody 2NSP23. Virtually equal Nsp9 affinities were previously measured with 2NSP23 and 2NSP90 by ELISA tests [12], and equivalent profiles of NMR peak loss and attenuation were observed in Nsp9 titrations with each nanobody (datra in S1 Table), which makes reasonable the comparison with the recently reported Nsp9-2NSP23 crystal structure [21]. Differently from the 4:4 stoichiometry compatible with the previous [12] and the present NMR-based evidence, the crystallographic result exhibits an overall 2:2 stoichiometry, with two homologous epitopes on the Nsp9 dimer surface, each including residues from β4 and β5 strands of either monomer [21]. The NMR-based and X-ray structures only share the epitope on β4 strand comprising W53. The X-ray complex, however, shows a rather profound conformational change of each Nsp9 monomer with respect to the geometry that has always been observed for the protein, also in rather distant viral species (Fig 5).

The torsion that swaps inwards the Nsp9 C-terminal helix, between the sheets β2-β3 and β4-β5 (Fig 5A and 5B), involves a further major rearrangement of the dimer geometry and interface, from parallel to antiparallel pairing of the monomers and contacting helices (Fig 5C and 5D). All this should bring about substantial chemical shift changes in the Nsp9 NMR spectrum that were never observed. Instead, upon titration with 2NSP90 and 2NSP23, the map of the backbone amides of Nsp9 exhibited progressive loss of ~ 65% of the peaks (data in S1 Table) because of an intermediate exchange, on the NMR chemical shift scale, between free and nanobody-bound protein. A similar pattern also affects the spectrum of isolated Nsp9. Small chemical shift variations could be observed only for a few NH cross peaks out of those surviving the titrations, in particular at N- and C-terminals (A8, L9, R111 and Q113) [12]. This can be reconciled with the intermediate exchange pattern that elsewhere progressively cancels out most of the Nsp9 NMR signals if local high mobility allows for fast chemical shift averaging, which looks quite conceivable at the chain termini. The fitting of the Nsp9 chemical shift changes at A8 and Q113 upon titration with 2NSP90 gave statistically reliable Hill coefficients, precisely 2.9±0.6 ($p = 1.5 \times 10^{-3}$) and 4.0±0.5 ($p = 3.7 \times 10^{-5}$), respectively [12], which are consistent with the binding of 3 or 4 nanobody units to Nsp9 complex. The Nsp9 cross-peak loss upon nanobody addition was reproduced also by the authors of the recent Nsp9-2NSP23 structure [21] who attributed the result to the increased size of the complex. Albeit not necessarily easy to assign, their 54 kDa molecular complex should be observable in HSQC spectra (S5 Fig) [51]. Only the broadening from an exchange process occurring at an intermediate rate on the chemical shift scale can justify the gradual signal loss that is observed when titrating Nsp9 with nanobodies. A conformational transition of Nsp9 from the canonical fold to the one observed in the Nsp9-2NSP23 structure [21], that would take place at such an intermediate rate, would simultaneously broaden all the signals of each protein molecule undergoing the conformational transition, leading to an overall spectrum attenuated by an extent related to the added amount of titrant, prior to final cancellation. Contrarily, the experiments always show sparse resonance losses and different attenuation rates of the surviving peaks, that are much more consistent with localized, non-uniform exchange interactions such as those expected with increasing concentration of the titrant nanobodies. As pointed out by the authors of the Nsp9-2NSP23 structure [21], the crystallization conditions were "non-

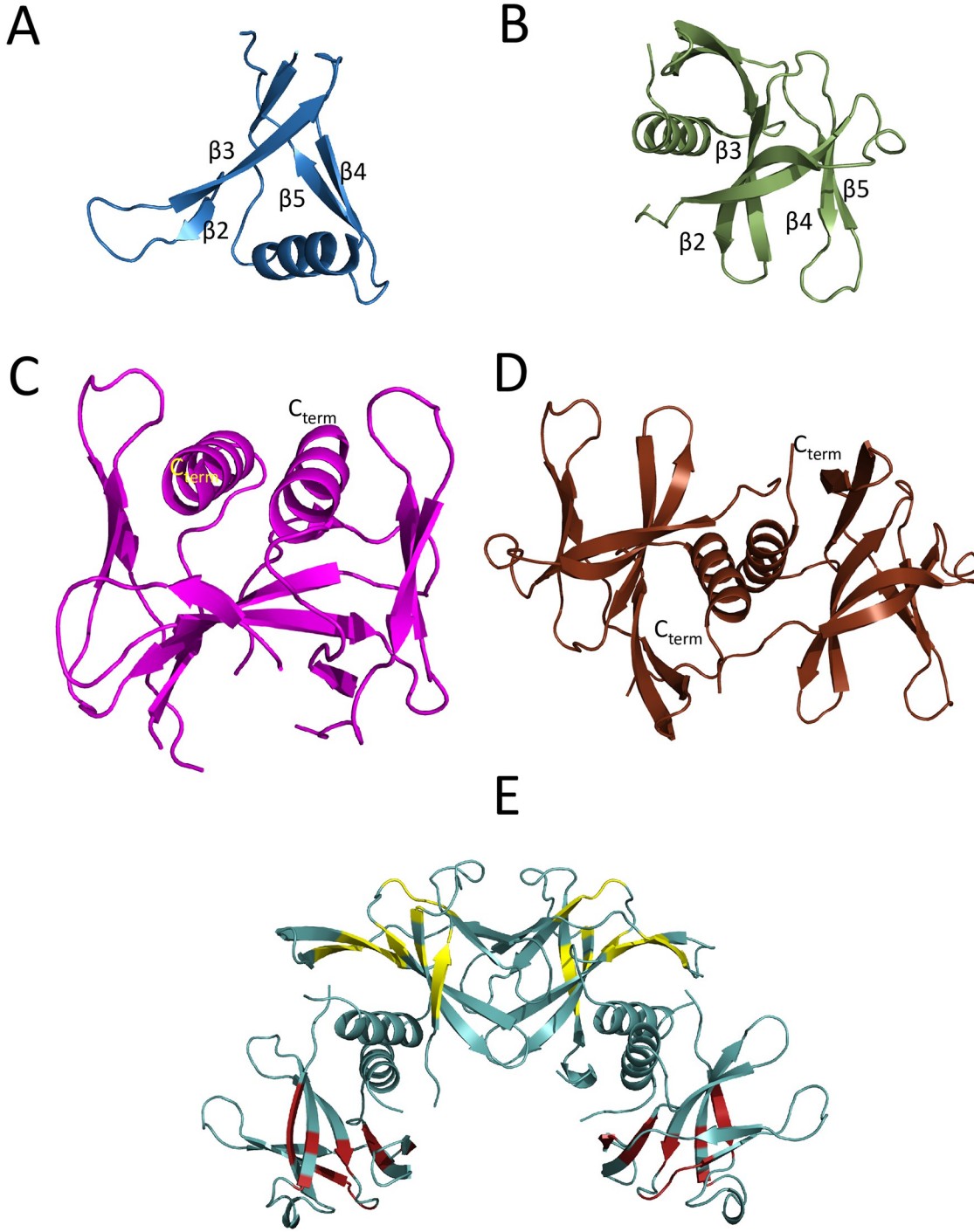

**Fig 5. Structural options of Nsp9 antigen.** (A) The Nsp9 monomer occurring in the crystal structure of the complex with 2NSP23 [21] (PDB ID: 8dqu). The C-terminal α-helix is oriented inwards between β2β3 and β4β5 sheets. (B) The Nsp9 monomer in the canonical dimer structure [42] (PDB ID: 6wxd). The C-terminal α-helix is external with respect to the β-strand segments. (C) The Nsp9 dimer in the crystal structure of the complex with 2NSP23 [21] (PDB ID: 8dqu). The two subunits show an antiparallel pairing as evident from the opposite orientations of the C-terminal α-helices. (D) The Nsp9 canonical dimer[42] (PDB ID: 6wxd). The two subunits display a staggered parallel pairing that can be appreciated from the orientations of the C-terminal α-helices. (E) The Nsp9 tetramer [39] (PDB ID: 7bwq). The location of the epitopes involved in the interaction with 2NSP90 and 2NSP23 is highlighted in yellow on the subunits involved in the dimer-dimer interface (also reported in Fig 4). The distribution of the same epitopes is likely to extend to the distal subunits from the tetramerization interface (dark red highlighting) to avoid the crowding of 4 nanobody molecules that are simultaneously complexed to the tetramerization interface proximal subunits. All structures were drawn with PyMol (Schroedinger LLC).

physiological" and surely different from the mild ones employed for the NMR characterizations in solution. Therefore extreme concentration and ionic strength conditions may have induced an alternative folding option for Nsp9 in the complex with the nanobody. As for the complex stoichiometry in solution, although the available NMR evidence support a 4:4 composition, the simultaneous occurrence of an alternative 2:2 stoichiometry can not be ruled out *a priori*. Multiple stoichiometries could be the consequence of Nsp9 oligomeric distribution. All the mapped epitopes, however, could hardly fit in complexes with low stoichiometries, but a crowding problem should also arise in the 4:4 assemblies, if nanobody binding occurs simultaneously only on the epitope arrangement of Fig 4. It is therefore quite likely that the epitope distribution over the Nsp9 tetramer surface also includes the two distal subunits with respect to the tetramerization interface (Fig 5E). This would also improve the dynamic averaging of the nanobody-Nsp9 interaction, consistently with the experimentally observed NMR equivalence of all Nsp9 subunits throughout nanobody titrations.

## Conclusions

We have presented here further experimental and computational evidence for the characterization of the interaction between SARS-CoV-2 Nsp9 and nanobody 2NSP90. Our results confirm the previously identified Nsp9 epitopes in the interaction with 2NSP90 [12], favoring a 4:4 stoichiometry of the protein-nanobody complex. The comparison of our evidence in solution with the recently reported crystal structure of the Nsp9 complex with 2NSP23 [21], a nanobody showing the same affinity and interaction profile as 2NSP90 [12], remarks differences in the involved structure of Nsp9 and its interactions within the complex. Only the epitope centered at W53 in the crystal structure of the complex is also observed in solution, where the conformation of Nsp9 retains the same fold as the isolated protein. Moreover the 2:2 stoichiometry of the Nsp9-2NSP23 crystal structure may be unsuitable to accommodate all the mapped epitopes on the dimer surface. The distribution of the mapped epitopes is probably spread over all the tetramer subunits to limit the crowding (Fig 5E). Together with the previous evidence on the oligomers in isolated Nsp9 [18], the NMR data support a 4:4 stoichiometry, but cannot rule out the simultaneous occurrence of a 2:2 assembly of the complex. Whichever the involved stoichiometries, using effective nanobodies to target the Nsps of coronaviruses may prove a very convenient and valuable strategy to develop antiviral therapies considering the low mutation propensity of several Nsps [3,15,16]. In particular, for SARS-CoV-2, the mapping of humoral immunity identified antibodies against non-structural proteins to be associated with the survival of critical patients [17]. For Nsp9, two such epitopes were found at segments 56–60 and 86–90 [17]. Interestingly, the epitopes of the Nsp9 tetramer previously described [12] and here confirmed include residues 86 (e2) and 87-88-89 (e4) (Figs 4 and 5E). The fragment 56–60 is displaced with respect to the epitope 50–53 (e3) mapped on the Nsp9 tetramer, but insists on the same β4 strand of the protein, thus identifying a significant immunogenic determinant. Based on the described evidence, we have successfully set up an experimental strategy to test *in vivo* the efficacy of the anti-Nsp9 nanobodies, as recently reported [52]. Along the same lines as the testing strategy, antiviral drugs may be prepared for a direct application of our results. The successful inhibition of viral replication based on Nsp9 targeting [52] also shows that nanobodies such as 2NSP23 and 2NSP90, raised against the Nsp9 mutant with all cysteines replaced by serines [12], recognize the natural Nsp9 sequence devoid of additional N-terminal extensions, and that the possible conformational bias arising from the presence of such extensions in the variants used to elicit the immune response or to study the selected nanobody interactions [12] is negligible or absent.

## Supporting information

**S1 Fig. ESR spectra overlay.** The ESR spectra of A) TEMPOL alone (black trace), TEMPOL with SARS-CoV-2 Nsp9 + 2NSP90 (red trace), and B) TEMPOL with SARS-CoV-2 Nsp9 + 2NSP23 (green trace), TEMPOL with SARS-CoV-2 Nsp9 (blue trace) superimpose very well, confirming the invariance of the nitroxide dynamic regime in the presence of the proteins and hence the absence of specific tight interactions, consistently with the $\tau_c$ values reported in the main text that were calculated from signal spacings and amplitudes [18–20]. The trace overlay was split into two panels to avoid graphic crowding. The concentration of any species was always 10 μM.
(PDF)

**S2 Fig. Paramagnetic perturbation of Nsp9 spectra.** $^{15}N$–$^{1}H$ HSCQ spectra overlay of labeled SARS-CoV-2 Nsp9 without (A) and with (B) unlabeled 2NSP90, in the absence (black contours) and presence (red contours) of TEMPOL paramagnetic perturbation. The assignments of the backbone NHs (and W53 side chain) [26] are also reported. Data collection was carried out at 298 K on 18 μM Nsp9 alone or with 5.6 μM 2NSP90, before and after addition of 1 mM TEMPOL, with a relaxation delay of 0.3 s, according to the PENELOP protocol for off-equilibrium acquisitions [18]. The contour plot pairs with and without TEMPOL are always drawn at the same vertical scale, whereas a two-fold scale increment is applied to panel B data (with 2NSP90) compared to panel A data (without 2NSP90).
(PDF)

**S3 Fig. Normalized attenuation ($A_N$) pattern of Nsp9 – 2NSP90 system.** Overlay plot of the $A_N$ values obtained from $^{1}H$-$^{15}N$ HSQC spectra of 18 μM SARS-CoV-2 Nsp9 in the absence (A) and presence (B) of 5.6 μM 2NSP90 as determined by 1.0 mM Tempol at 298 K with a relaxation delay of 0.3 s (blue, nonequilibrium condition) and 3 s (red, equilibrium condition). The data for segments and residues 1–7, 20, 36, 59, 87, 96–106, and the relative abscissa points are not reported because of the absence of the corresponding signals from the spectra. The locations of prolines (devoid of NH) are also skipped on the abscissa axis.
(PDF)

**S4 Fig. Nsp9 normalized attenuation differences.** Bar plot of {$A_N$[off] − $A_N$[eq]} differences highlighting the locations of the Type I pattern (green bars) and Type II pattern (purple bars) for 18 μM Nsp9 alone (A) and in the presence of 5.6 μM 2NSP90 (B). Same plot as Fig 3 of main text without expansion truncation. Refer to Fig 3 caption for color code and other information.
(PDF)

**S5 Fig. High molecular weight protein NMR.** $^{15}N$-$^{1}H$ HSQC spectrum U ($^{15}N$, $^{13}C$) 70% $^{2}H$ EMILIN1 C1q domain, a 52 kDa homotrimer. The spectrum was obtained at 17.6 T (750 MHz $^{1}H$ frequency) and 310 K. The most crowded spectral region shown in the map benefits from 70% $^{2}H$ labeling, but a lower resolution spectrum is also observed with the fully protonated sample.
(PDF)

**S1 Table. Peak loss and attenuation in $^{15}N$-$^{1}H$ HSQC spectra of Nsp9 titrations with nanobodies.**
(PDF)

**S1 File. Normalized attenuation ($A_N$) values and relative error definitions.**
(PDF)

## Acknowledgments

The Core Technology Platform of New York University Abu Dhabi is acknowledged for the use of the instrumental facilities.

## Author Contributions

**Conceptualization:** Gennaro Esposito, Federico Fogolari, Piergiorgio Percipalle.

**Data curation:** Gennaro Esposito, Yamanappa Hunashal, Mathias Percipalle, Federico Fogolari.

**Formal analysis:** Gennaro Esposito, Yamanappa Hunashal, Mathias Percipalle, Federico Fogolari.

**Funding acquisition:** Gennaro Esposito, Piergiorgio Percipalle.

**Investigation:** Gennaro Esposito, Yamanappa Hunashal, Mathias Percipalle, Federico Fogolari.

**Methodology:** Gennaro Esposito, Yamanappa Hunashal, Federico Fogolari.

**Resources:** Tomas Venit, Ainars Leonchiks, Fabio Piano.

**Writing – original draft:** Gennaro Esposito.

**Writing – review & editing:** Gennaro Esposito, Yamanappa Hunashal, Mathias Percipalle, Federico Fogolari, Tomas Venit, Ainars Leonchiks, Kristin C. Gunsalus, Fabio Piano, Piergiorgio Percipalle.

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
