## [Decision Letter · Decision Letter 0]

2 Nov 2023

PONE-D-23-21838Assessing nanobody interaction with SARS-CoV-2 Nsp9PLOS ONE

Dear Dr. Esposito,

Thank you for submitting your manuscript to PLOS ONE. After careful consideration, we feel that it has merit but does not fully meet PLOS ONE’s publication criteria as it currently stands. Therefore, we invite you to submit a revised version of the manuscript that addresses the points raised during the review process.

We look forward to receiving your revised manuscript.

Kind regards,

Krupakar Parthasarathy, PhD

Academic Editor

PLOS ONE

Journal Requirements:

2. We note that you have a patent relating to material pertinent to this article. Please provide an amended statement of Competing Interests to declare this patent (with details including name and number), along with any other relevant declarations relating to employment, consultancy, patents, products in development or modified products etc. Please confirm that this does not alter your adherence to all PLOS ONE policies on sharing data and materials, as detailed online in our guide for authors http://journals.plos.org/plosone/s/competing-interests by including the following statement: "This does not alter our adherence to  PLOS ONE policies on sharing data and materials.” If there are restrictions on sharing of data and/or materials, please state these. Please note that we cannot proceed with consideration of your article until this information has been declared.

“This work was supported by Tamkeen under the NYU Abu Dhabi Research Institute Award to the NYUAD Center for Genomics and Systems Biology (ADHPG-CGSB). The Core Technology Platform of New York University Abu Dhabi is acknowledged for the use of the instrumental facilities.”

“The authors received no specific funding for this work. YH and MP worked on the NYUAD institutional grant of GE. TV worked on the NYUAD institutional of PP.”

Additional Editor Comments (if provided):

Major revision Needed.

Reviewers' comments:

Reviewer's Responses to Questions

**Comments to the Author**

1. Is the manuscript technically sound, and do the data support the conclusions?

Reviewer #1: No

Reviewer #2: Partly

Reviewer #3: Yes

Reviewer #4: Yes

2. Has the statistical analysis been performed appropriately and rigorously? 

Reviewer #1: No

Reviewer #2: Yes

Reviewer #3: Yes

Reviewer #4: N/A

3. Have the authors made all data underlying the findings in their manuscript fully available?

Reviewer #1: No

Reviewer #2: No

Reviewer #3: No

Reviewer #4: No

4. Is the manuscript presented in an intelligible fashion and written in standard English?

Reviewer #1: No

Reviewer #2: Yes

Reviewer #3: Yes

Reviewer #4: Yes

5. Review Comments to the Author

Reviewer #1: 1.Introduction should contain about expirement question without writting about result and conclusion

2.References format should be maintained as per journal in materials and methods

3.Justify your results listed in methodology .

4.Discussion should be comparative with the obtained results and avaiable data about the study

5.Conclusion must include furthur research and progression that can be done with the above research along with their applications.

Reviewer #2: The paper builds on the work already done by this group, Adv Biol (Weinh)

. 2021 Dec;5(12):e2101113. doi: 10.1002/adbi.202101113. Epub 2021 Oct 27..

The PENELOP technique itself seems sound (has been published by the same group - no citations so far though - in well known and reputable journals (Analytical Chemistry ref 18 and PCCP ref 19) and the analysis of the technique in ref 19 in particular is good. The conclusions they draw from the technique wrt exposed surfaces and buried surfaces of the protein in the presence of the nanobody are compelling. Conclusions related to exchange dynamics are a bit less so - ref 19 says this is a reasonable explanation but stops short of saying this is the only explanation - but it seems to only be a passing mention in the paper. Needs to be validated

The Figure 1 presents off-equilibrium HSQC spectra alone. Having the equilibrium HSQC spectra in addition to (or instead of) off-equilibrium spectra would be more useful to emphasise the conclusions they draw in that section (they say that they present raw A_N values in figure S2 of the supplementary information but I can't see this figure)

While the methodology is nifty, the conclusions can further validated . In both the papers, the antiviral activity has not been demonstrated. If the nanobodies work across strains again, the potency not demonstrated.

Reviewer #3: This study is basically an extension of a previous study by the same authors (Esposito et al, Adv Biol 5(12):2101113; 2021) where the nanobody was isolated and characterised by 2D NMR spectroscopy and chemical shift analysis.

Results of this work includes a very limited amount of new experimental data (PENELOP protocol, previously established, based on NMR paramagnetic perturbation) in addition to new molecular dynamics simulations. The authors confirmed their previously published NMR data in terms of stoichiometry and epitope mapping.

The very comprehensive discussion section contains a comparison to a recently published structure by X-Ray crystallography (Pan et al, PLoS One 18(4): e0283194; 2023) revealing significant differences, in particular no significant structural rearrangement is observed in the present study compared to the crystal structure.

Overall, the experimental execution is solid, and the conclusions supported by the data, however I have significant reservations about the novelty of this manuscript. A very limited amount of new data is presented, and the data basically just confirm a model that is already published in the author’s earlier work.

At the very least, a detailed comparison between the model presented and the published crystal structure is required and a set of Nsp9 mutants should be designed to either confirm or exclude the model presented in this study.

Also, have effort being made to crystallise the presented Nsp9-nanobody complex? This is not mentioned in the manuscript.

Reviewer #4: In the work titled “Assessing nanobody interaction with SARS-CoV-2 Nsp9”, Esposito et al combined PENELOP, a paramagnetic perturbation methodology, and molecular dynamics simulations to further dissect the interaction between nsp9 and nanobody 2NSP90. NMPylation/RNAylation of Nsp9 and their roles in SARS-CoV-2 capping, which is being studied extensively. Besides the medical usage, nanobody can also be an extra tool for nsp9 studies. Mapping the binding sites and characterizing the binding features of nsp9 – nanobody is essential. Some concerns need to be addressed before being considered for acceptance.

Major:

1. The Discussion and Conclusion sections are too long, which makes it hard to catch the main point. Importantly, it’s difficult to see the new findings from this study.

2. Page 11 lines 14 – 20: please provide a schematic to illustrate the Type I and Type II patterns for a broader reader; Figure 3: Please provide a rotated structure view to show the analogous epitope ensemble; S1 figure doesn’t match the results on page 11 line 22. Only TEMPOL alone and TEMPOL + nsp9 + 2NSP90 are shown.

3. Please make the NMR data publicly available.

4. For discussing the increased Type II pattern of e1, the authors used “translating into an additional interaction exchange” on page 15 – line 5. But what does it mean? Even as stated on page 19 line 19 “loose enough to preserve the Nsp9 surface accessibility for the paramagnetic probe”, it still doesn’t explain the increased Type-II pattern. Is it because the nanobody disrupts the interdimer domain?

Minor:

1. Page 3 line 14: please delete “the most interesting”; lines 16 – 20 can be rephrased into one shorter sentence.

2. Page 4 line 16: “contributing less than 1% of total mutations” might be because nsp7 – 10 are small.

3. Page 6 line 12: it’s better to write “H2O/D2O 95/5 (vol/vol)”; Line 18: “Microliter aliquots” of what? Also, “1 mM nitroxide”.

4. Page 7 line 12: delete “see”. In general, please delete “see” from all the figure citations.

5. Page 9 line 5: what are the parameters? line 6: Which one has more numbers in the slightly different number scenario? Line 20: “k-th”.

6. Page 10 line 5: “0.150 M” is used here, which is a different format from other places, such as page 9 – line 10. Please keep one format for it; Line 10: α-carbons (also page 16 – line 16); Line 11: “the dissociating subunits”; Line 13: delete “the simulation”; Line 15: what differences?

7. The citation format on page 9 – lines 15/18, and page 10 – lines 4/12, is not right, please correct them and check the whole manuscript.

8. Page 11 lines 1 – 5: please rephrase the sentences here to make it clear.

9. Page 12 lines 5 – 7: repeated words as in Figure 1 legend can be found in these sentences. Please rephrase to reduce redundancy.

10. Figure 2: please indicate Figure 2 is truncated at the beginning of the legend to avoid confusion; Please indicate V85 in the figure; And why do the authors use different x-axis scales?

11. Page 13 line 7: it’s nicer to show the fragment/residues sequentially; line 20: missing a comma.

12. Page 14 line 2: “locations”; line 14: delete “see”; line 18: “histograms” is not the correct word here. One should use “bar graph”.

13. Figure S3: in the legend, should it be “Same plot as Fig. 2 of main text”?

14. Page 15 line 3: “increasing”; line 4: which data support “stressed by the low nanobody/Nsp9 ratio”?

15. Page 18 lines 10 – 11: please rewrite “An intermediate exchange in fact broadens up to merging with the baseline the NMR signal linewidths of the exchanging species, thereby preventing detection.” It’s difficult to understand now.

6. PLOS authors have the option to publish the peer review history of their article (what does this mean?). If published, this will include your full peer review and any attached files.

Reviewer #1: No

Reviewer #2: No

Reviewer #3: No

Reviewer #4: No

---

## [Author Response · Author response to Decision Letter 0]

25 Dec 2023

Reviewer 1

Reviewer 1:1.Introduction should contain about expirement question without writting about result and conclusion

All references to results and conclusions were eliminated from Introduction.

Reviewer 1: 2.References format should be maintained as per journal in materials and methods

The references reported twice in the text with proper and erroneous formats were corrected.

Reviewer 1: 3.Justify your results listed in methodology .

There is no methodology section listing our results in the main text, nor in Supplementary Information. We do not understand what the reviewer means.

Reviewer 1: 4.Discussion should be comparative with the obtained results and avaiable data about the study

The Discussion section the reviewer read in the first version of the manuscript was divided into two parts addressing the Nsp9 complex stoichiometry and the alternative structure recently reported by Pan and colleagues (PLoS ONE 2023;18(4): e0283194). Both subsections already consisted of comparisons of different stoichiometries and conformational options with the proposed composition and structural arrangement. These characteristics have been retained in the current version of the manuscript. Therefore the reviewer requests are implemented. 

Reviewer 1: 5.Conclusion must include furthur research and progression that can be done with the above research along with their applications.

We appreciate the suggestion of the reviewer. We added to the conclusions of the revised manuscript additional statements concerning the development of the research for possible applications.

 

Reviewer 2

Reviewer 2: The PENELOP technique itself seems sound (has been published by the same group - no citations so far though - in well known and reputable journals (Analytical Chemistry ref 18 and PCCP ref 19) and the analysis of the technique in ref 19 in particular is good. The conclusions they draw from the technique wrt exposed surfaces and buried surfaces of the protein in the presence of the nanobody are compelling. Conclusions related to exchange dynamics are a bit less so - ref 19 says this is a reasonable explanation but stops short of saying this is the only explanation - but it seems to only be a passing mention in the paper. Needs to be validated

We thank the reviewer for appreciating our work on paramagnetic perturbation of protein spectra. Based on the reviewer comment, that appreciation is only partial, however. The reviewer in fact seems to cast doubts on the interpretation of Type II pattern of PENELOP method when remarking the absence, in our previous paper, of an explanation other than slow-to-intermediate exchange dynamics. This statement is not accompanied by any of the possible alternative explanations the reviewer implicitly hints at and/or seems to glimpse, but a lapidary “Needs to be validated”. 

The works dealing with the subject (listed as refs. 18, 19, 20) were published in reputable journals after passing regular peer review processes and, within the context of the discussed applications, no objection concerning possible alternative explanations was ever raised. 

Reference 19, the one specifically cited by the reviewer, describes at length the basis of our interpretation providing both an independent experimental reproduction of the exchange dynamics mechanism (see Fig. S6 in Supplementary Information and page 6252 of reference 19), and an extensive validation of the results obtained from the proposed interpretation of equilibrium and off-equilibrium paramagnetic attenuation. The validation consists of independent and systematic measurements of relaxation dispersion and T1��at different spin-lock field strengths of the same molecular group signals that were considered for equilibrium and off-equilibrium paramagnetic perturbation. Therefore our conclusions related to exchange dynamics have already been thoroughly validated. 

Reviewer 2: The Figure 1 presents off-equilibrium HSQC spectra alone. Having the equilibrium HSQC spectra in addition to (or instead of) off-equilibrium spectra would be more useful to emphasise the conclusions they draw in that section (they say that they present raw A_N values in figure S2 of the supplementary information but I can't see this figure)

Figure 1 (renumbered Figure 2 in the revised version to comply with the requests of another reviewer) has been modified according to the reviewer suggestion and the comparison of the equilibrium HSQC maps in the absence ad presence of 2NSP90 is now reported. The corresponding comparison under non-equilibrium conditions is shown in Supplementary Information (Figure S2) to avoid extending further the manuscript (again to comply with the request from another reviewer). Upon resetting the HSQC overlays to prepare the new figure, we realized that the previously reported superpositions had been erroneously selected by inadvertently swapping the correct spectral maps with unrelated ones occurring in the same experiment directory. The mistake has been corrected in Figure 1 (and Figure S2), along with the relative main text comment to the same figure. The revision also enabled correcting erroneously reported information on residues 86 and 87. In addition, the reviewer mentions a missing Supplementary Information figure, namely Figure S2 (renumbered as Figure S3 in the revised version and reported below with the corresponding caption). We apologize for the inconvenience. We do not understand why that figure was missing. In the manuscript downloaded from the journal site, the final link to the Supplementary Information correctly retrieved the file and the figures therein. We hope the revised manuscript version will not present the problem highlighted by the reviewer.

Reviewer 2: While the methodology is nifty, the conclusions can further validated . In both the papers, the antiviral activity has not been demonstrated. If the nanobodies work across strains again, the potency not demonstrated.

The reviewer furtherly mentions validation, this time concerning the antiviral activity of the nanobodies. The current manuscript only deals with the in-vitro study of the interaction between Nsp9 and one of the previously reported nanobodies (2NSP90), much like the previously published works on the subject, i.e. Esposito et al., Adv. Biol. 2021; 5 (12):2101113; and Pan et al., PLoS ONE 2023;18(4): e0283194, numbered 12 and 21 in our reference list. The conclusions of all the previous in-vitro characterizations suggested a possible application of the tested nanobodies for antiviral therapy. The manuscript under consideration reaches the same conclusions using additional, different evidence. However, as reported in the conclusion section of the revised version, a successful in-vivo test of the antiviral properties of the addressed nanobodies was recently obtained. Therefore the antiviral activity of the nanobodies has indeed been demonstrated.

 

Reviewer 3

Reviewer 3: This study is basically an extension of a previous study by the same authors (Esposito et al, Adv Biol 5(12):2101113; 2021) where the nanobody was isolated and characterised by 2D NMR spectroscopy and chemical shift analysis.

Results of this work includes a very limited amount of new experimental data (PENELOP protocol, previously established, based on NMR paramagnetic perturbation) in addition to new molecular dynamics simulations. The authors confirmed their previously published NMR data in terms of stoichiometry and epitope mapping.

The very comprehensive discussion section contains a comparison to a recently published structure by X-Ray crystallography (Pan et al, PLoS One 18(4): e0283194; 2023) revealing significant differences, in particular no significant structural rearrangement is observed in the present study compared to the crystal structure.

Overall, the experimental execution is solid, and the conclusions supported by the data, however I have significant reservations about the novelty of this manuscript. A very limited amount of new data is presented, and the data basically just confirm a model that is already published in the author’s earlier work.

We thank the reviewer for appreciating the solidity of our work. We do not agree with the reviewer’s concern about the limited novelty of our findings. It is in fact necessary to take into account the circumstances that prompted us to carry out the measurements that are presented in the manuscript. In march 2022 we were contacted by Dr. Littler, the corresponding author of the paper by Pan et al. (PLoS ONE 2023;18(4): e0283194), as mentioned in the former cover letter of our submission. He proposed a collaboration based on their crystallographic results. We replied we could accept the collaboration if a section comparing the different results could be included in the manuscript. We also sent a draft for that section but no longer heard from Dr. Littler until his paper came out. As it can be easily verified, the work by Littler’s group barely mentions our previous conclusions. The same work also reports an NMR titration for which a coarsely wrong interpretation is proposed that obscures the already published and discussed significance of our (and their) titration, thereby ruling out any discussion of the experimental observations in solution. Therefore, the manuscript which is now being considered is the product of further efforts of our group to provide alternative and additional evidence challenging our previous results in comparison with those deriving from an authoritative technique such as X-rays. As a matter of fact, the new evidence support our former interpretation. 

As well known, in the presence of a dispute, PLoS ONE policy entails submitting the disputing manuscript to the author of the disputed paper. This procedure has been accepted and applied for our manuscript too. It is worth to point out that the irreprensible policy PLoS ONE adopted with our manuscript was not applied for the PLoS ONE manuscript by Dr. Littler and colleagues who expressed and studied the complex of Nsp9 with a nanobody that was first published by us reaching different conclusions. We do not see any fault of the journal nor the reviewers because the difference with respect to the previous results reads softened or misinterpreted in the paper by Dr. Littler and colleagues. Indeed, one cannot expect the Editor or the reviewers to examine in detail all the quoted literature.

Reviewer 3: At the very least, a detailed comparison between the model presented and the published crystal structure is required and a set of Nsp9 mutants should be designed to either confirm or exclude the model presented in this study.

Also, have effort being made to crystallise the presented Nsp9-nanobody complex? This is not mentioned in the manuscript.

Our manuscript reports the description of the differences between the model we propose and the crystal structure recently published by Pan and coworkers (PLoS ONE 2023;18(4): e0283194). In their paper those authors show a quite detailed comparison between their peculiar monomer and dimer structures and the typical Nsp9 monomer and dimer arrangements found in any previous crystal structure of the protein. The tetrameric assembly that we propose to be targeted by 2NSP90 and 2NSP23 corresponds to the crystallographic structure published by Zhang and colleagues (Mol. Biomed. 2020; 1:5, reference 39 in our manuscript) which exhibits both inter-helix and inter-�-sheet interfaces. The details of that assembly are found in the original work by Zhang et al. (ref. 39) and in references 40-45 of our list. In our manuscript the most salient features are however comparatively highlighted. To limit the length of the work, as requested by other reviewers, we omit additional some details that can be recovered from the quoted literature.

Except for the mutant with Cys14, Cys23 and Cys73 replaced by serines, we did not test other Nsp9 variants. A systematic investigation involving Nsp9 mutants could provide evidence on the mapped epitope composition and uniqueness, although the repertoire of the collected nanobodies is quite large. The 136 earlier-reported nanobody sequences that were selected by ELISA test (see our reference 12) bear 40 diverse CDR3 which may well extend the range of recognized motifs. On the other hand, using a different methodology to verify our former conclusions appeared a preferable option as it could allow a more focused test and hence a more direct reply to the result published by Pan and colleagues. 

We did no attempt to crystallize the Nsp9-2NSP23 or Nsp9-2NSP90 complexes. Pan and colleagues report quite harsh crystallization conditions that were defined “non-physiological” and might have determined the profound structural change undergone by Nsp9. 

 

Reviewer 4

Reviewer 4: In the work titled “Assessing nanobody interaction with SARS-CoV-2 Nsp9”, Esposito et al combined PENELOP, a paramagnetic perturbation methodology, and molecular dynamics simulations to further dissect the interaction between nsp9 and nanobody 2NSP90. NMPylation/RNAylation of Nsp9 and their roles in SARS-CoV-2 capping, which is being studied extensively. Besides the medical usage, nanobody can also be an extra tool for nsp9 studies. Mapping the binding sites and characterizing the binding features of nsp9 – nanobody is essential. Some concerns need to be addressed before being considered for acceptance.

We thank very much the reviewer for the meticulous reading of our manuscript and the precious suggestions. We are also pleased the reviewer shares our view on the relevance of Nsp9 epitope mapping.

Reviewer 4: Major:

1. The Discussion and Conclusion sections are too long, which makes it hard to catch the main point. Importantly, it’s difficult to see the new findings from this study.

2. Page 11 lines 14 – 20: please provide a schematic to illustrate the Type I and Type II patterns for a broader reader; Figure 3: Please provide a rotated structure view to show the analogous epitope ensemble; S1 figure doesn’t match the results on page 11 line 22. Only TEMPOL alone and TEMPOL + nsp9 + 2NSP90 are shown.

3. Please make the NMR data publicly available.

4. For discussing the increased Type II pattern of e1, the authors used “translating into an additional interaction exchange” on page 15 – line 5. But what does it mean? Even as stated on page 19 line 19 “loose enough to preserve the Nsp9 surface accessibility for the paramagnetic probe”, it still doesn’t explain the increased Type-II pattern. Is it because the nanobody disrupts the interdimer domain?

1.The Conclusions and the Discussion sections were partially shortened. In fact other reviewers asked to discuss in detail the comparison between our results and the literature evidence which we did essentially in the Discussion section. As for the novelty of the findings, it is necessary to take into account the circumstances that prompted us to publish the results that are presented in the manuscript. In march 2022 we were contacted by Dr. Littler, the corresponding author of the paper by Pan et al. (PLoS ONE 2023;18(4): e0283194), as mentioned in the former cover letter of our submission. He proposed a collaboration based on their crystallographic results. We replied we could accept the collaboration if a section comparing the different results could be included in the manuscript. We also sent a draft for that section but no longer heard from Dr. Littler until his paper came out. As it can be easily verified, the work by Littler’s group barely mentions our previous conclusions. The same work also reports an NMR titration for which a coarsely wrong interpretation is proposed that obscures the already published and discussed significance of our (and their) titration, thereby ruling out any discussion of the experimental observations in solution. Therefore, the manuscript which is now being considered is the product of further efforts of our group to provide alternative and additional evidence challenging our previous results in comparison with those deriving from an authoritative technique such as X-rays. As a matter of fact, the new evidence support our former interpretation.

2.Schematics are not allowed by PLoS ONE style. Therefore a new figure (Figure 1) has been in

---

## [Decision Letter · Decision Letter 1]

18 Apr 2024

PONE-D-23-21838R1Assessing nanobody interaction with SARS-CoV-2 Nsp9PLOS ONE

Dear Dr. Esposito,

Thank you for submitting your manuscript to PLOS ONE, and my honest apologies for the delay in getting back to you with an answer. As you can see, all reviewers agree with its acceptance but one of them underlines several important issues that must be constructively discussed. Therefore, we invite you to submit a revised version of the manuscript by Jun 02 2024 11:59PM that convincingly addresses the points raised by the reviewer.

We look forward to receiving your revised manuscript.

Kind regards,

Maria Gasset, Ph.D.

Academic Editor

PLOS ONE

Journal Requirements:

Reviewers' comments:

Reviewer's Responses to Questions

**Comments to the Author**

1. If the authors have adequately addressed your comments raised in a previous round of review and you feel that this manuscript is now acceptable for publication, you may indicate that here to bypass the “Comments to the Author” section, enter your conflict of interest statement in the “Confidential to Editor” section, and submit your "Accept" recommendation.

Reviewer #3: All comments have been addressed

Reviewer #4: All comments have been addressed

Reviewer #5: All comments have been addressed

2. Is the manuscript technically sound, and do the data support the conclusions?

Reviewer #3: Yes

Reviewer #4: Yes

Reviewer #5: Partly

3. Has the statistical analysis been performed appropriately and rigorously? 

Reviewer #3: Yes

Reviewer #4: Yes

Reviewer #5: I Don't Know

4. Have the authors made all data underlying the findings in their manuscript fully available?

Reviewer #3: Yes

Reviewer #4: Yes

Reviewer #5: Yes

5. Is the manuscript presented in an intelligible fashion and written in standard English?

Reviewer #3: Yes

Reviewer #4: Yes

Reviewer #5: Yes

6. Review Comments to the Author

Reviewer #3: All comments have been addressed.

All comments have been addressed.

All comments have been addressed.

Reviewer #4: (No Response)

Reviewer #5: My understanding is that there are several issues to consider. Coronaviral Nsp9 is predominantly a homodimer in solution, but its active form within the RTC is NiRAN-associated. However other cellular functions have been reported for Nsp9 which may utilise its homodimeric form. Secondly a free N-terminal amine on Asn-1 is now known to be an essential feature of this protein and is required for RNAylation activity. Many prior papers have recombinant “stubs” such as the authors current GAMG sequence, or my own GPG sequence for constructs used in Littler et al. 2020. This complicates the ability to draw physiologically relevant conclusions from stub-Nsp9 work.

In Esposito et al. 2021 Adv Biol the authors immunized llama with a triSer-Nsp9 mutant (C14S, C23S, C73S). I believe their own NMR suggests significant structural changes for this mutation as compared to wild type. Cysteines are unusual in that have both hydrophobic and polar characteristics. When they are buried within a hydrophobic pocket a serine mutant can be disruptive and structurally and in some instances an alanine mutation might work better. Indeed, most of the peaks lost correspond the changes seen in 8DQU. Any antibodies with this as immunogen could act to enforce whatever the triSer conformation is, which would be simple enough to crystalise as a control and check.

I cannot comment on 2nsp90 as it’s CDR3 loop is distinct from the nanobody I published upon. I drew my sequences for both antibodies from page 19 of the authors supplementary material ADBI-5-2101113-s001.pdf. If major differences in behaviour are being observed can the authors quickly confirm this sequence is correct.

In my hands 2nsp23 and 2nsp90 certainly bind Nsp9 but direct affinity measurements via SPR and ITC appeared poor for an antibody. Hence the absence of this data in Pan et al. It remains remotely possible these antibodies behave differently when expressed via mammalian systems. I had assumed the poor binding to be due to the requirement of the structural change leading to slower than normal on-rates. It would take significant work to assess whether this is true, I see this as a potentially interesting mechanism by which to inhibit a protein and may aid the inhibitory potential of the LNP system deployed by this group.

I contacted the authors previously about inclusion on the Pan et al manuscript, they were more inclined to want to include discussion on the tetramer form of Nsp9 than I felt comfortable with. Especially as some in the community now even doubt the physiological relevance of the homodimer. I am confident the structure observed is what was present in the crystals and have discussed this with minimal conclusions. I have retained the line requested about the potential for structural influences upon crystallisation but did not to want to include one about Nsp9-tetramerization in the NMR data as I cannot speak to it. I am yet to be convinced by the tetramer interface.

The authors are convinced by their proposed mechanism. I don’t have any experience with PENELOP interpretation. With enough funds and interest, it would be trivial to obtain new crystal structures from a different condition of Nsp9:2Nsp90 or Nsp9:2Nsp23. Via an independent group if necessary.

I had hoped one of these antibodies would be useful to pull-down Nsp9 within cells or to aid crystallisation for drug screens. Their slow on-rate for wild-type Nsp9 and enforced structural change upon binding limit their use as such tools. If ture, this is useful information for the community. My understanding of certain techniques, and lack of understanding in PENELOP, will naturally bias my conclusions so I will leave them to stand as written. I will happily reinterpret the data however if I or others find clearer structural data to the contrary that is easily interpretable.

7. PLOS authors have the option to publish the peer review history of their article (what does this mean?). If published, this will include your full peer review and any attached files.

Reviewer #3: No

Reviewer #4: No

Reviewer #5: **Yes: **Dr Dene Littler

---

## [Author Response · Author response to Decision Letter 1]

27 Apr 2024

Reviewer #5

We addressed all criticisms, requests and suggestions that were raised in the first round of the reviewers’ scrutiny. This definitely helped to improve our manuscript, as acknowledged by all reviewers. We were a bit surprised on reading the further comments of Dr. Littler. The newly raised issues concern subjects that sometime can be included in the text as specific caveats to further improve it and adhere to the reviewer’s requests. In the following, all new concerns are examined in detail.

My understanding is that there are several issues to consider. Coronaviral Nsp9 is predominantly a homodimer in solution, but its active form within the RTC is NiRAN-associated. However other cellular functions have been reported for Nsp9 which may utilise its homodimeric form. Secondly a free N-terminal amine on Asn-1 is now known to be an essential feature of this protein and is required for RNAylation activity. Many prior papers have recombinant “stubs” such as the authors current GAMG sequence, or my own GPG sequence for constructs used in Littler et al. 2020. This complicates the ability to draw physiologically relevant conclusions from stub-Nsp9 work.

The predominance of the Nsp9 homodimeric state in solution is clearly stressed in our manuscript, along with the monomeric state requirement for the formation of the viral replication and transcription complex (RTC). Following to the reviewer suggestion, the further information regarding the role of homodimers for other cellular functions was added in the Discussion (first subsection, pg. 18, line 10). 

As for the role of the ”stubs” we do not understand the point raised by the reviewer. Surely a free Asn1, i.e. without “stub”, is necessary for the nucleotidylation of the N-terminal amine, a well-established event characterizing the role of Nsp9 in the RTC (Wang et al., Nucl. Ac. Res. 2021). However our work deals with the interaction of nanobodies with Nsp9. This interaction could be affected by the presence of polypeptide extentions preceding the N-terminal residue of Nsp9, because the local conformation might change thereby potentially biasing the interaction with the nanobody. There are, however, two arguments that Dr. Littler should consider. First, a few Nsp9 crystallographic structures have been obtained with constructs including “stubs” (e.g. 6w4b, 1uw7, 6wxd) that apparently did not introduce major conformational variations with respect to the sequence devoid of N-terminal extensions (e.g. 1qz8, 2J97, 3ee7, 7bwq, 2J98). Most meaningfully, all the X-ray structures exhibit a typically disordered N-terminal conformation involving the first 3-6 residues of the Nsp9 sequence, except some of those bearing a (declared) “stub” that apparently attenuates the N-terminal conformational dispersion. In solution, however, the presence of N-terminal elongations does not restrict the Nsp9 N-terminal mobility which was always observed to entail an intermediate conformational exchange with local loss of resonance detectability (Buchko et al., Biomol. NMR Assign. 2021; Dudas et al., Biomol. NMR Assign. 2021; El-Kamand et al., Proteins 2022). We observed the same mobility and resonance loss pattern with either short (a single Met residue) and longer (GlyAlaMetGly) N-terminal extensions (Esposito et al., Adv. Biol. 2021). The second, much more stringent argument stems from the results we recently reported (Venit et al., BioRxiv 2023), as mentioned in ref. 52 of the manuscript under consideration. The anti-Nsp9 nanobody 2NSP23 inhibits viral replication by targeting Nsp9 in living cells. Hence, in spite of possible effects of the “stub” on the conformational mobility of the N-terminal region of the Nsp9 variants used to elicit the immune response or to study the selected nanobody interactions, the latter species targets successfully the natural Nsp9 sequence in SARS-CoV-2 infected cells. A statement concerning the possible bias introduced by the “stub” was anyway added at the end of the Conclusions section of the manuscript (pg. 25, lines 11-17). 

In Esposito et al. 2021 Adv Biol the authors immunized llama with a triSer-Nsp9 mutant (C14S, C23S, C73S). I believe their own NMR suggests significant structural changes for this mutation as compared to wild type. Cysteines are unusual in that have both hydrophobic and polar characteristics. When they are buried within a hydrophobic pocket a serine mutant can be disruptive and structurally and in some instances an alanine mutation might work better. Indeed, most of the peaks lost correspond the changes seen in 8DQU. Any antibodies with this as immunogen could act to enforce whatever the triSer conformation is, which would be simple enough to crystalise as a control and check.

The statement of the reviewer about the “significant structural changes” of triSer-Nsp9 based on our NMR evidence has no foundation but the personal belief of the reviewer. And even the argument on the disruptive character of the Cys->Ser mutations when in hydrophobic pockets seems controversial. According to PAM120 replacement matrix, the Cys->Ser replacement is neutral (scoring 0), whereas the Cys->Ala one is unfavorable (scoring -3). Different scorings are assigned, instead, in BLOSUM62, with nearly equivalent estimates for Ala (0) and Ser(-1). At any rate, the three cysteines of Nsp9 are close to the surface and not buried in hydrophobic pockets. Moreover, we wish to point out that our conclusions were based on a detailed comparison of the NMR spectra of wild-type and triSer variants of Nsp9. As discussed in our former publication (Esposito et al., Adv. Biol. 2021), the HSQC maps of wild-type and triSer Nsp9 were indeed observed to differ by the loss, in the mutant spectra, of resonances that match the inter-dimer contacts of the tetrameric structures of SARS-CoV-2 Nsp9 (Zhang et al., Mol. Biomed. 2020) and SARS-CoV Nsp9 (Miknis et al., J. Virol. 2009). The missing signals are indicated in Fig. 2E of our Adv. Biol. 2021 paper with wild-type assignments obtained from BMRB. On the other hand, the patterns of the observable NH correlations in the HSQC maps of wild-type and triSer Nsp9, albeit not identical, were essentially similar and consistent with limited chemical shift deviations plausibly reflecting solvation differences. The intermediate exchange dynamics of dimerization and tetramerization were also proven comparing the diffusion coefficients of the two protein variants and hen egg white lysozyme. After expressing his interpretation on the comparison between wild-type Nsp9 and triSer mutant, with “supporting” considerations on Ser substitution for Cys, Dr. Littler adds that ”most of the peaks lost correspond the changes seen in 8DQU”. From his writing, it is not clear which set of peak loss he refers to. As 8dqu is the structure of the Nsp9 complex with 2NSP23, the concerned peak loss should be the one occurring upon titration of wild-type Nsp9 with the nanobody. This peak loss is described in great detail in S1 Table of Supplementary Information of the manuscript under consideration. As explained at length in the text, that progressive loss of backbone NH resonances maps the different Nsp9 epitope regions as well as the Nsp9 inter-subunit contacts (within the dimer and the tetramer). We never observed changes of chemical shifts upon titration with 2NSP23 and 2NSP90. We only observed progressive signal losses on increasing the titrant nanobody concentration. Such a pattern is not at all compatible with the profound structural change of Nsp9 depicted by 8dqu. If the Nsp9 subunits complexed with 2NSP23 had undergone the transition from the canonical folding to the one observed in 8dqu, a simultaneous change of the vast majority of the chemical shifts should have occurred which, under conditions of intermediate exchange, would have led to simultaneous broadening of all resonances. Therefore the statement of Dr. Littler is against the experimental NMR evidence, even the one he himself published (Pan et al., PlosOne 2023) and misinterpreted by attributing the signal loss to the size of the Nsp9-2NSP23 complex. Paradoxically, Pan and colleagues conclude their NMR analysis stating that their “results are broadly consistent with previously reported titration experiments and confirm a central role of Trp-53 within the epitope [25]”, where reference [25] is Esposito et. al., Adv. Biol. 2021. Why then “the widespread broadening” of the HSQC peaks was not deemed to match the changes seen in 8dqu, whereas such a matching is envisaged only for the peak loss we report? 

I cannot comment on 2nsp90 as it’s CDR3 loop is distinct from the nanobody I published upon. I drew my sequences for both antibodies from page 19 of the authors supplementary material ADBI-5-2101113-s001.pdf. If major differences in behaviour are being observed can the authors quickly confirm this sequence is correct.

The sequences on page 19 of Adv. Biol. 5, 2101113, 2021 Supplementary Information are correct and the corresponding nanobodies do not exhibit major behavior difference in the Nsp9 titration experiments (S1 Table of Supplementary Information of the manuscript under consideration). 

In my hands 2nsp23 and 2nsp90 certainly bind Nsp9 but direct affinity measurements via SPR and ITC appeared poor for an antibody. Hence the absence of this data in Pan et al. It remains remotely possible these antibodies behave differently when expressed via mammalian systems. I had assumed the poor binding to be due to the requirement of the structural change leading to slower than normal on-rates. It would take significant work to assess whether this is true, I see this as a potentially interesting mechanism by which to inhibit a protein and may aid the inhibitory potential of the LNP system deployed by this group.

We do not have any clue about a different behavior of nanobodies expressed in mammalian systems. The ELISA test of 2NSP23 and 2NSP90 binding to Nsp9 worked satisfactorily in our hands, with nanobodies expressed in E.coli WK6. In addition, the two mentioned nanobodies exhibited high expression yields and were therefore selected for further testing. The corresponding data are reported in Tables S1 and S2 of the Supplementary Information in Esposito et al., Adv. Biol. 5, 2101113, 2021. In our opinion, any further comment relating the poor nanobody performance to “the structural change leading to slower than normal on-rates” appears superfluous. 

I contacted the authors previously about inclusion on the Pan et al manuscript, they were more inclined to want to include discussion on the tetramer form of Nsp9 than I felt comfortable with. Especially as some in the community now even doubt the physiological relevance of the homodimer. I am confident the structure observed is what was present in the crystals and have discussed this with minimal conclusions. I have retained the line requested about the potential for structural influences upon crystallisation but did not to want to include one about Nsp9-tetramerization in the NMR data as I cannot speak to it. I am yet to be convinced by the tetramer interface.

Dr. Littler’s version omits to mention that, besides the crystallographic results, the paper by Pan et al. also reports the NMR experiment of Nsp9 titration with 2NSP23 nanobody he and his colleagues interpreted ignoring the picture emerging from the same experiment we had published two years before (Esposito et al., Adv. Biol. 2021). In addition, Pan and colleagues provided a wrong explanation for the resonance loss they too observed, and cited only the minor feature of the Trp53 presence in Nsp9 epitope to support a picture of converging evidence, in spite of totally diverging contexts of interpretation. 

The authors are convinced by their proposed mechanism. I don’t have any experience with PENELOP interpretation. With enough funds and interest, it would be trivial to obtain new crystal structures from a different condition of Nsp9:2Nsp90 or Nsp9:2Nsp23. Via an independent group if necessary.

I had hoped one of these antibodies would be useful to pull-down Nsp9 within cells or to aid crystallisation for drug screens. Their slow on-rate for wild-type Nsp9 and enforced structural change upon binding limit their use as such tools. If ture, this is useful information for the community. My understanding of certain techniques, and lack of understanding in PENELOP, will naturally bias my conclusions so I will leave them to stand as written. I will happily reinterpret the data however if I or others find clearer structural data to the contrary that is easily interpretable.

We appreciate the availability of Dr. Littler to reconsider his interpretation in the presence of contrary and easily interpretable evidence. We also thank him for kindly leaving “as written” the PENELOP conclusions he does not understand.

Dr. Littler’s poor understanding of PENELOP cannot however condition the acceptance of our manuscript, especially if other reviewers, presumably more competent in that area, had already commented as follows: 

i)“The PENELOP technique itself seems sound (has been published by the same group - no citations so far though - in well known and reputable journals (Analytical Chemistry ref 18 and PCCP ref 19) and the analysis of the technique in ref 19 in particular is good.”

ii) “Overall, the experimental execution is solid, and the conclusions supported by the data”. 

Nonetheless, Dr. Littler asks for further experimental verifications through new crystallographic determinations ”Via an independent group if necessary”. 

In our opinion this request is particularly excessive in the presence of compelling evidence demonstrating i) the impossibility of interpreting the NMR titration data of Nsp9 solutions with both 2NSP23 and 2NSP90 nanobodies in terms of the extensive folding modification observed in the Nsp9-2NSP23 crystal structure reported by Pan and colleagues (8dqu); ii) the successful inhibition of the viral replication in SARS-CoV-2 infected cells due to the nanobody 2NSP23 which was designed to target Nsp9, i.e. an essential component of the viral replication and transcription complex. 

We do not intend to cast any doubt on the Nsp9-2NSP23 complex structure obtained by Dr. Littler and colleagues, but sometimes, especially for small proteins, crystal structures faithfully report only the occurrence of artifacts enforced by crystallization conditions.

---

## [Editor Report · Decision Letter 2]

2 May 2024

Assessing nanobody interaction with SARS-CoV-2 Nsp9

PONE-D-23-21838R2

Dear Dr. Gennaro Esposito,

First of all, my personal apologies for the extent of the reviewing process and the acknowledge for the last effort. We’re pleased to inform you that your manuscript has been judged scientifically suitable for publication and will be formally accepted for publication once it meets all outstanding technical requirements.

Kind regards,

Maria Gasset, Ph.D.

Academic Editor

PLOS ONE
---

## [Editor Report · Acceptance letter]

8 May 2024

PONE-D-23-21838R2 

PLOS ONE

Dear Dr. Esposito, 

I'm pleased to inform you that your manuscript has been deemed suitable for publication in PLOS ONE. Congratulations! Your manuscript is now being handed over to our production team.

Kind regards, 

on behalf of

Dr. Maria Gasset 

Academic Editor

PLOS ONE